# Hydrogen storage with gravel and pipes in lakes and reservoirs

Julian David Hunt [1,2] ✉, Andreas Nascimento[3], Oldrich Joel Romero [4], Behnam Zakeri [2,5], Jakub Jurasz[6], Paweł B. Dąbek [6], Tomasz Strzyżewski[7], Bojan Đurin[8], Walter Leal Filho [9], Marcos Aurélio Vasconcelos Freitas[10] & Yoshihide Wada [1]

Climate change is projected to have substantial economic, social, and environmental impacts worldwide. Currently, the leading solutions for hydrogen storage are in salt caverns, and depleted natural gas reservoirs. However, the required geological formations are limited to certain regions. To increase alternatives for hydrogen storage, this paper proposes storing hydrogen in pipes filled with gravel in lakes, hydropower, and pumped hydro storage reservoirs. Hydrogen is insoluble in water, non-toxic, and does not threaten aquatic life. Results show the levelized cost of hydrogen storage to be 0.17 USD kg$^{-1}$ at 200 m depth, which is competitive with other large scale hydrogen storage options. Storing hydrogen in lakes, hydropower, and pumped hydro storage reservoirs increases the alternatives for storing hydrogen and might support the development of a hydrogen economy in the future. The global potential for hydrogen storage in reservoirs and lakes is 3 and 12 PWh, respectively. Hydrogen storage in lakes and reservoirs can support the development of a hydrogen economy in the future by providing abundant and cheap hydrogen storage.

The green hydrogen economy has the potential to replace fossil fuels as the primary source of energy for transportation, industrial processes, and electricity generation[1]. Green hydrogen is an energy carrier produced from renewable sources such as wind, solar, and hydropower through electrolysis that can help address the challenges posed by climate change[2]. However, there are issues associated with implementing a hydrogen economy, including the high cost of producing, storing, and transporting hydrogen[3]. In the USA, renewable generation in 2023 was 21%, with solar power expected to increase by 30% between 2023 and 2024[4]. The renewable electricity generation share in the European Union (EU) energy mix has risen from 28.4% in 2015 on average to 46.8% in 2024[5]. If this trend is maintained, the EU could generate 100% renewable energy in 2050. However, for this to be achieved, energy storage technologies, such as batteries, pumped storage, and hydrogen, are required[6]. One of the main challenges associated with storing solar power seasonally during the summer is recovering it during winter[7]. Possible seasonal energy storage technologies are hydrogen[8], ammonia[9], pumped storage[10], compressed air energy storage[11], gravity energy storage[12], and biomass[13]. According to McKinsey & Company[14], the demand for gray hydrogen was 100 million tons in 2023, and it is estimated that 600 million tons of green hydrogen will be required by 2050 to achieve net zero emissions.

[1]Biological and Environmental Science and Engineering Division, King Abdullah University of Science and Technology, Thuwal, Makkah, Saudi Arabia. [2]International Institute of Applied Systems Analysis (IIASA), Laxenburg, Niedaöstareich, Austria. [3]Federal University of Itajubá, Itajubá, Minas Gerais, Brazil. [4]Federal University of Espírito Santo, São Mateus, Espírito Santo, Brazil. [5]Institute for Data, Energy, and Sustainability (IDEaS), Vienna University of Economics and Business (WU), Vienna, Austria. [6]Wrocław University of Science and Technology, Wrocław, Województwo dolnośląskie, Poland. [7]Institute of Meteorology and Water Management, National Research Institute, Warsaw, Masovian, Poland. [8]Department of Civil Engineering, University North, Koprivnica, Koprivnica-Križevci, Croatia. [9]Faculty of Life Sciences, Hamburg University of Applied Sciences, Hamburg, Lower Saxony, Germany. [10]Federal University of Rio de Janeiro, Rio de Janeiro, Rio de Janeiro, Brazil. ✉e-mail: julian.hunt@kaust.edu.sa

There are several large-scale seasonal hydrogen storage technologies, and the most discussed for the future green hydrogen economy are compressed hydrogen in depleted natural gas reservoirs and salt caverns[15]. However, appropriate gas reservoirs and salt caverns are not widely available everywhere[16]. Another storage option that has gained substantial attention is underwater compressed gas storage[17]. Research on underwater compressed gas energy storage has steadily grown in recent years and may be classified as flexible or rigid. Flexible gas storage accumulators store gases by changing the volume of the accumulator and are based on composite materials that have been explored and demonstrated in academia and industry. However, they still remain unreliable in a harsh and complicated marine environment[18]. The Energy Bag company created a balloon-shaped flexible energy bag and tested it in saltwater at the European Marine Energy Center in Orkney[19]. Hydrostor Inc. evaluated two types of flexible air accumulators after learning from the commonly used lift bags in maritime engineering. By the end of 2015, they had successfully built and managed the world's first grid-connected underwater compressed-air energy storage (UWCAES) system on Toronto Island[20].

Rigid gas storage accumulators store gases by replacing the water in the tank to maintain a constant pressure. As energy carriers are generally non-polar (oil, natural gas, hydrogen, compressed air), they are insoluble in water and do not require a membrane to separate them[21]. Marine constructions made of reinforced concrete or steel are reliable and feasible. However, because the gas storage accumulator must withstand current flow, high salinity, high pressure, scouring, and other complex effects over long periods, a series of problems, such as concrete cracking and reinforcement corrosion, may occur[22]. Offshore engineering projects utilizing steel concrete structures, such as concrete support structures for offshore wind turbines, sea crossing bridges, undersea immersed tunnels, and offshore floating platforms, have evolved in recent years[23]. Proposals similar to the one explored in this paper have been made to store hydrogen in pipelines filled with desert sand in the deep ocean[8,24]. More details on underwater gas storage accumulators can be seen in ref. [17].

As discussed above, the literature on underwater gas storage accumulators focuses mostly on storing gas in the deep ocean, with only a few studies looking at lakes and reservoirs, focusing on compressed air energy storage. The main contributions and objectives of this work are: (i) investigate the possibility of storing hydrogen in lakes and reservoirs. Storing hydrogen in lakes and reservoirs offers greater flexibility in terms of storage location and other benefits discussed in the paper. (ii) The proposed storage container in this paper consists of high-density polyethylene (HDPE) pipelines filled with gravel. (iii) The costs of the system and (iv) the world's potential for the technology are explored.

## Results

Hydrogen storage in lakes and reservoirs, as described in the method section, is possible due to the low solubility of hydrogen in water. If the pressure in the tank is 20 bar, the solubility is 0.0004% of the molar mass of $H_2$ in water[25] (Fig. 1a). This results in an insignificant loss of hydrogen with the operation of the storage tank. Another requirement is that the pressure of the underwater hydrogen tank should always be the same as its surroundings. This way, the pipeline storing the gas can be relatively thin and cheap. Figure 1b presents the density of hydrogen, water, and gravel at 15 °C and different depths. As can be seen, the greater the depth, the more hydrogen is stored in the tank. For example, at a depth of 100 m, the density is 0.838 kg m$^{-3}$, and at 1000 m, the density is 7.864 kg m$^{-3}$. cc presents the change in hydrogen density with temperature and pressure[26]. An increase in temperature from 10 to 20 °C at 10 bar reduces the hydrogen density from 0.853 kg m$^{-3}$ to 0.823 kg m$^{-3}$, respectively. That is a reduction of 0.36% in density per 1 °C. Figure 1d shows that the maximum hydrogen

volume that can be stored in the tank without submerging is 62.53% of the total volume at a 200 m depth, including the voids between the gravel particles. Figure 1e presents the energy density variation with the tank's depth, which varies from 27.9 kWh m$^{-3}$ at 100 m of depth to 261.9 kWh m$^{-3}$ at 1000 m of depth, assuming 33.3 kWh kg$^{-1}$ of $H_2$. This shows that the deeper the hydrogen tank is located, the more hydrogen and energy can be stored.

Clean gravel from mine waste with granularity above 5 cm was selected as ballast to increase the hydrogen tank's weight and keep it in the bottom of the reservoir. This is because using material with lower granularity, such as sand, would result in challenges in inserting the $H_2$ in the tank due to the capillary effect of the water and permeability hysteresis when removing the $H_2$ from the tank. The cost of the gravel from mine waste varies depending on the availability of mining activity near the hydrogen storage location and the transport distance from the source. Finding other uses for mine waste is convenient as it reduces the costs of discarding it. The main characteristic of gravel that impacts hydrogen storage costs is its density. The larger the density, the less gravel mass is required, and thus, more hydrogen can be stored in the tank. The gravel porosity does not impact the total hydrogen stored in the tank. The pipeline could be filled with concrete, but concrete degrades with time (gravel does not degrade with time), and the cost of concrete is around 30 USD ton$^{-1}$, substantially increasing the cost of hydrogen storage. The cost to connect the future hydrogen grid to the hydrogen tank in the lake or reservoir is not considered. Table 1 presents a cost estimate for hydrogen storage in lakes or reservoirs 200 m deep and at 15 °C, including cost uncertainties. The estimated levelized cost of hydrogen storage is 0.17 USD kg$^{-1}$. Figure 1f presents the hydrogen tank components investment cost distribution. Figure 1g presents the initial investment cost estimate for the hydrogen tank components and the cost uncertainty considered in this study. Figure 1h presents the hydrogen storage investment cost variation with depth in lakes and reservoirs. As the figure shows, the cost of hydrogen storage reduces substantially with the tank's depth. Table 2 compares the proposed $H_2$ storage in lakes and reservoirs with other $H_2$ storage options, showing it is competitive with other large-scale hydrogen storage alternatives.

### Hydrogen storage in Oroville Lake, California

This theoretical case study looks into the possibility of storing hydrogen in California's Oroville Lake. Oroville Lake is a hydroelectric reservoir in Butte County constructed by the Oroville Dam impounding the Feather River. The reservoir covers 64.75 km$^2$ and holds 4.3 km$^3$ of water. The hydroelectric project includes three 132 MW turbines and three 141 MW pump turbines, totaling 819 MW[27]. When there is excess electricity, the Hyatt Powerplant may pump water back into Lake Oroville. At maximum output, the pump turbines at Hyatt can raise 159 m$^3$ s$^{-1}$ into Lake Oroville, while the six turbines utilize a total flow of 480 m$^3$ s$^{-1}$ [27]. The Oroville reservoir has a maximum depth of 210 m and an area of 466,300 m$^2$ at the bottom of the reservoir, where the hydrogen can be stored at depths greater than 200 m[28]. We could not find measurements of the temperature on the bottom of the Oroville reservoir, so we assumed it to be 8 °C, which is equal to the average daily temperature of January (the coldest month). One pipe with 10 m diameter and 100 m long occupies a 1000 m$^2$ area and can store 4836 m$^3$ of hydrogen at bar 20.6 at 1.65 kg m$^{-3}$ density. This is equivalent to 7983 kg of hydrogen, equivalent to 186 MWh, assuming an electricity generation efficiency of 70%. As the Oroville reservoir can accommodate 462 hydrogen tanks, it can store 3,679,650 kg of $H_2$ or 86 GWh of electricity. This potential could be increased by increasing the diameter of the hydrogen tanks.

Figure 2a presents the solar generation in Oroville, the projected electricity demand for solar generation, and the energy storage with hydrogen. Solar generation was estimated using the renewables.ninja website[29], assuming the coordinate is close to the Oroville

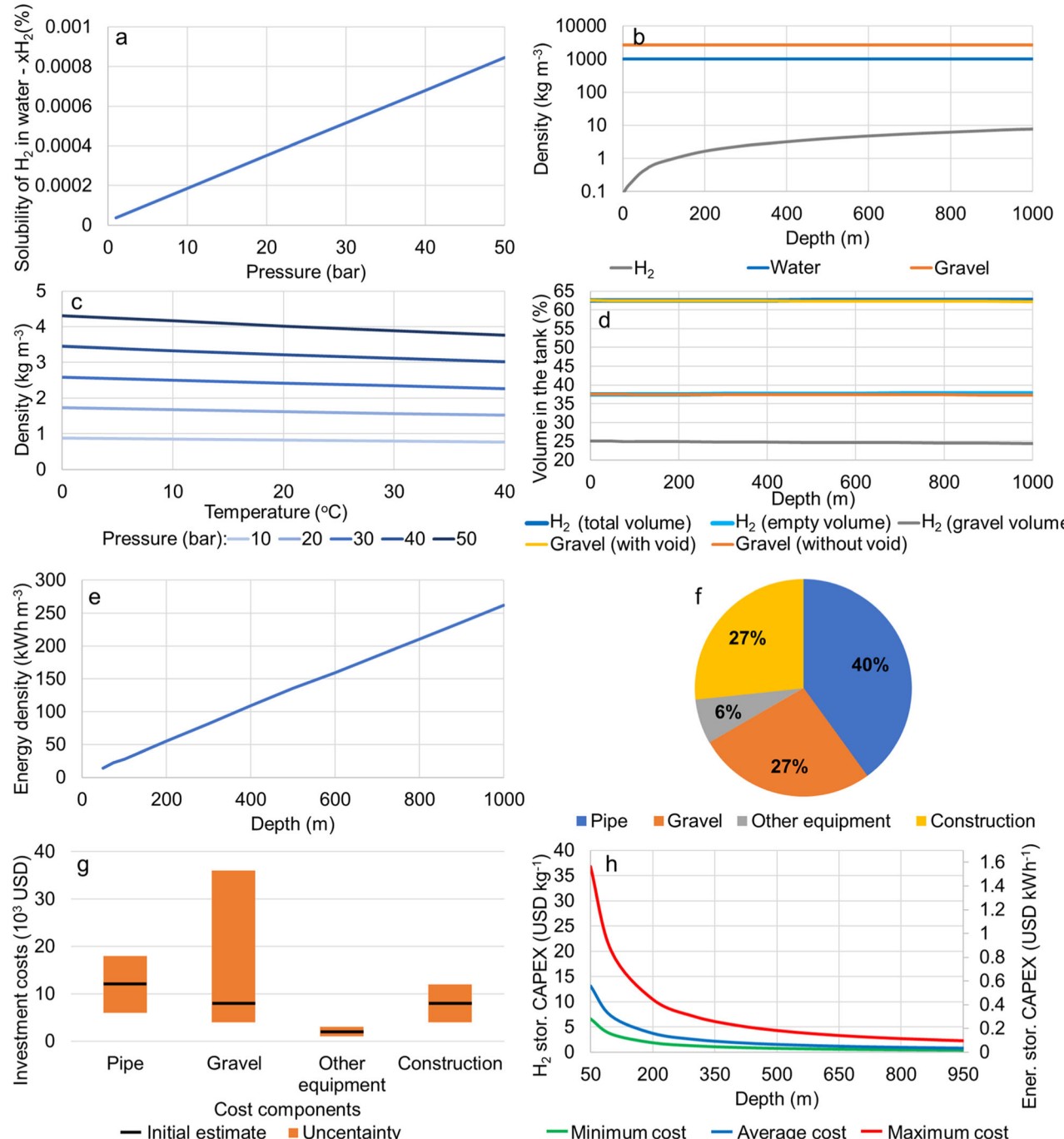

**Fig. 1 | Characteristics of hydrogen storage at different depths. a** solubility of hydrogen in water at different pressures, (**b**) hydrogen, water, and gravel density at 15 °C[26], (**c**) hydrogen density change with temperature and pressure[26], (**d**) hydrogen and gravel volume required to avoid the tank from submerging, (**d**) solubility of hydrogen in water at different pressures, (**e**) energy density of the hydrogen tank at different depths, (**f**) hydrogen tank component investment cost distribution, (**g**) investment cost uncertainty considered for hydrogen storage tanks, (**h**) hydrogen and energy storage investment cost variation with depth in lakes and reservoirs.

dam (39°32′20″N 121°29′08″W), with no tracking and a tilt of 35% facing south. The solar plant has a 400 MW installed capacity and a capacity factor of 19% to supply a fixed energy demand of 70 MW. The storage system consists of batteries (to store the solar power in daily cycles), electrolysis, and fuel cells with 330 MW installed capacity and 86 GWh storage capacity. The storage system manages to supply the electricity demand, but 14.6% of the solar energy is curtailed. Figure 2b presents the Oroville reservoir level[30] and solar generation. Figure 2c presents the Oroville reservoir level[30] and hydrogen density in the tank. The higher the reservoir level, the higher the hydrogen density and the more energy is stored in the tanks. It should be noted that the

hydrogen storage tanks should only be installed below the reservoir's dead storage capacity or the lake's minimum historical level. If the tank ends up above the water level, it cannot be used for hydrogen storage, and the tanks will reduce the water storage capacity of the reservoir or lake and their capacity to control drought and floods. Figure 2d presents the tank's energy and hydrogen volume storage capacity. A possible disadvantage of this system is that during dry years, solar generation is high, but the reservoir level might drop, reducing the hydrogen storage capacity. This disadvantageous match between storage potential and solar surplus might lead to solar generation curtailment in certain years.

**Table 1 | Capital cost estimate for a hydrogen storage tank 200 m deep and at 15 °C**

| Component | Cost description | Value |
|---|---|---|
| Pipe | HDPE pipe with 10 meters in diameter and 100 m long (7850 m³), extrapolating the costs in ref. 45. Depending on the difficulty of access to the reservoir, transport costs can double the cost of the pipeline. | 12,000 USD |
| Gravel | Gravel from mine waste is assumed to cost 1 USD per ton to fill a volume of 2.938 m³ with 2,666 kg m⁻³ density (around 8000 tonnes). Depending on the availability of gravel and transport distance, the cost can increase to 5 USD per ton. | 8000 USD |
| Other equipment | Valves and sensors. The costs can increase depending on the size of the tank. | 2000 USD |
| Construction | 50% of the equipment costs, as equipment costs are low and underwater construction costs are high. This cost can double due to the availability and accessibility of construction equipment and personnel. | 8000 USD |
| Investment costs | Total investment costs | 30,000 USD |
| Hydrogen storage capacity | The hydrogen storage capacity is 4836 m³ at 20.6 bar pressure and 1651 kg m⁻³ density. | 7983 kg |
| Hydrogen storage CAPEX | Capital investment (CAPEX) to store 7983 kg with a 30,000 USD investment cost. | 3.76 USD kg⁻¹ |
| Operation and maintenance costs | Assumed to be 5% of the investment costs per year. | 1500 USD y⁻¹ |
| Lifetime | A lifetime of 30 years. Note that the gravel has a much higher lifetime. | 30 years |
| Cycles per year | Two cycles per year. This includes the seasonal, monthly, weekly, and daily hydrogen storage cycles. | 3 cycles |
| Interest rate | Interest rate of 8%. | 8% |
| Levelized cost of hydrogen storage | Levelized cost of hydrogen storage (LCOS) | 0.17 USD kg⁻¹ |

**Table 2 | Comparison between $H_2$ storage in lakes and reservoirs with other $H_2$ storage options[46]**

| | Lakes and reservoirs | Salt caverns | Depleted gas fields | Rock caverns | Pressurized containers |
|---|---|---|---|---|---|
| Cycles per years | Seasonal, months, weeks | Months, weeks | Seasonal | Months, weeks | Daily |
| Capacity (tons) | 10–10,000 | 300–10,000 | 300–10,000 | 300–2500 | 0.1–2 |
| LCOS (USD kg⁻¹) | 0.17 | 0.11 | 1.07 | 0.23 | 0.17 |
| Availability | Limited | Limited | Limited | Limited | Not limited |

## Global potential

The global potential for storing hydrogen in lakes and reservoirs is shown in Fig. 3a. There are more than 1.43 million lakes[31] globally, but only 1760 lakes have met the specifications described in the methods section. This results in a total hydrogen storage capacity of 12 PWh, with the Caspian Sea representing more than half of this potential (6.4 PWh). If we exclude from the analysis the five largest lakes, the potential drops to 1.9 PWh. The other four lakes with the largest potential are Baikal (1.96 PWh), Tanganyika (1.57 PWh), Superior (1.02 PWh), and Malawi (0.65 PWh). An interactive map has been created in Fig. 3b so that the reader can better explore the potential for hydrogen storage in lakes and reservoirs[32]. Figure 3c presents the hydrogen storage cost vs potential curve per region. The Former Soviet Union has the highest hydrogen storage potential because of the Caspian Sea (8.83 PWh). This is followed by North America (2.61 PWh), Sub-Saharan Africa (2.60 PWh), Latin America and the Caribbean (0.324 PWh), Western Europe (0.167 PWh), Centrally Planned Asia and China (0.164 PWh), Other Pacific Asia (0.0624 PWh), Middle East and North Africa (0.0542 PWh), Pacific OECD (0.0375 PWh), South Asia (0.0256 PWh), and Eastern Europe (0.009 PWh). Figure 3c also includes the total global cost curve potential for hydrogen storage in lakes and reservoirs, together with a sensitivity analysis considering the cost uncertainties described in Fig. 1g, h. Future work focuses on improving the methodology for estimating the global potential for hydrogen storage in lakes and reservoirs as soon as bathymetric data on global lakes and reservoirs is available.

## Discussion

We investigated the possibility of storing hydrogen seasonally in the Oroville Reservoir, California. With a 0.46 km² area and 200 m depth, it can store 86 GWh of electricity with a levelized cost of hydrogen storage of 0.17 USD kg⁻¹, which is competitive with other large-scale hydrogen storage alternatives. This cost is similar to storing hydrogen in salt caverns and depleted natural gas reservoirs, The proximity to

deep reservoirs, lakes or salt caverns might dictate the viability of future hydrogen storage projects. Apart from storing hydrogen seasonally, empty tanks can also be used to store compressed air for daily and weekly energy storage cycles (Supplementary Table 3). The area required to generate solar power in Oroville with a 400 MW power plant is ~17.4 km². The area required to provide seasonal storage for solar generation (Fig. 2a) with hydrogen tanks in the Oroville reservoir is 0.46 km². Thus, the area required for storing hydrogen in lakes and reservoirs is 38 times smaller than that required to generate solar power.

Another advantage of having hydrogen storage in hydropower reservoirs is that the hydrogen can be produced using hydropower. This would reduce the electricity losses with transmission lines. Care should be taken to avoid heavy objects falling on and damaging the hydrogen tanks. This could be addressed by monitoring large vessels navigating over the tanks and by installing nets over the tanks to protect them from falling objects. The deep ocean has been the main focus for underwater compressed gas energy storage[17]. However, there are many advantages to storing gases in lakes and reservoirs. Supplementary Table 1 compares oceanic vs. reservoir underwater compressed gas energy storage. In locations with latitudes between 25 and 40 degrees, deep lakes and reservoirs have warm meromictic characteristics[33]. During the winter, the temperature of the reservoirs equals the average daily temperature of the coldest month, i.e., 6 to 15 °C. During the summer, the ambient temperature increases substantially, but below 40 m, the temperature remains at 6 to 15 °C. This is convenient for storing hydrogen and other gases in the bottom of the reservoir/lake (Supplementary Table 2). In addition, a freshwater air-conditioning system could be built parallel to the hydrogen storage plant. The cold water from the bottom of the reservoir could be used for cooling processes around the reservoir[34]. For example, it could be used to increase the efficiency of a hydrogen liquefaction plant. The cradle-to-gate life cycle analysis for producing HDPE using fossil fuels as materials and energy sources results in an overall $CO_2$ emission of

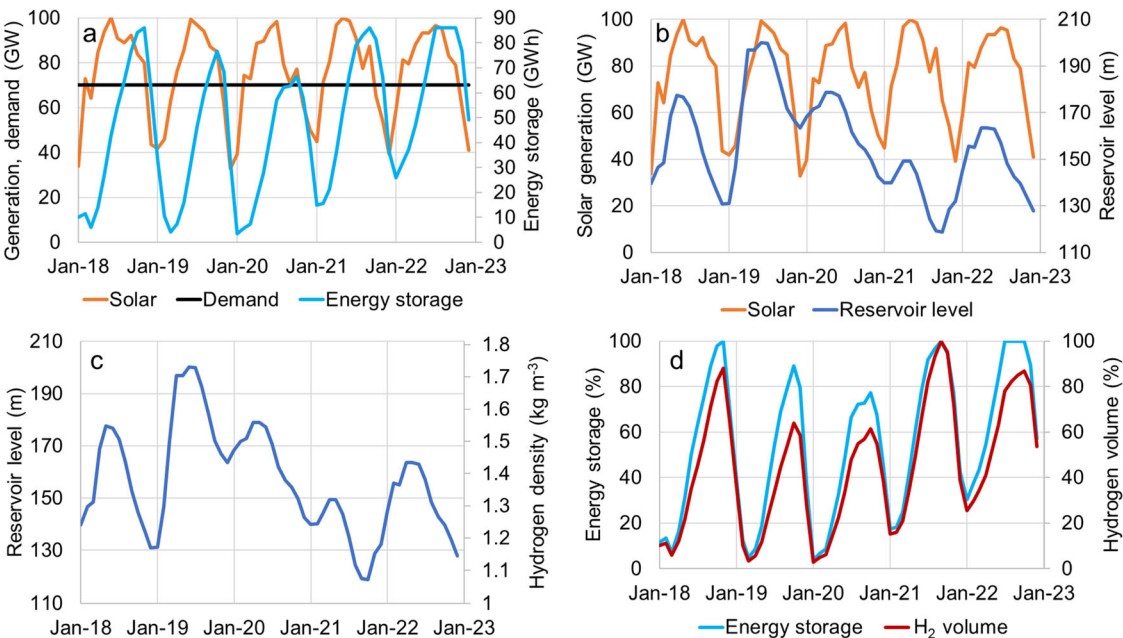

**Fig. 2 | Hydrogen storage in Oroville Lake California. a** solar generation in Oroville, projected electricity demand, energy storage with hydrogen from 2018 to 2023[29]. **b** Oroville reservoir level[30] and solar generation[29]. **c** Oroville reservoir level[30] and hydrogen density in the tank. **d** Energy storage and hydrogen volume in the tank.

1.6 kg per kg of HDPE[35]. Thus, a 300 tons tank would emit 480 tons of $CO_2$. A gas power plant to generate 186 MWh (the same energy stored as $H_2$ in the tank) would be 77 tons, assuming an emission of 413 kg of $CO_2$/MWh[36]. If the tank operates seasonally, it would take 6.2 Years for the tank to store the same amount of energy per $CO_2$ emission as a gas power plant. In the future, with the use of renewable electricity and $CO_2$ captured from the air, the production of the tank with the methanol-to-olefins process would result in negative emissions[37]. For instance, 85.7% of the tank's mass is carbon, which is equivalent to 257 tons of carbon, or 942 tons of $CO_2$ emissions captured and stored within the HDPE tank.

To estimate the global potential of hydrogen storage in underwater tanks, we have used databases for artificial reservoirs[38] and lakes[31]. Both contained information concerning the reservoir total area and volume, its location as well as average depth. Unfortunately, the total depth and detailed bathymetry were not available. Considering the limited data availability, we applied the procedure as follows: for each reservoir, the depth at which the tanks would be submerged is equal to the average reservoir depth, and the area that can be used for that purpose is equal to 10% of the total reservoir surface. Furthermore, we assume that the tanks are not spaced tightly and only 90% of this area can be used. It is assumed that the tanks are anchored to the bottom of the reservoir to prevent sideways movement. The databases were screened to select the reservoirs with an average depth of at least 30 m. This resulted in a list of 3403 man-made reservoirs and 1760 lakes. Note that 30 m depth is too shallow for building hydrogen storage tanks, as the pressure will be only approximately 4 bar and the cost for storing hydrogen will be high,' as shown in Fig. 1h. We added reservoirs with 30 m average depth to the global potential methodology because 10% of the total reservoir area of the reservoir might achieve 100 m depth or more, which is required for underwater hydrogen storage. However, this might not be the case for all reservoirs. This is a limitation of the methodology and data available. For a precise estimate of the hydrogen storage cost and potential of the reservoir, the reader needs to find bathymetric data of the reservoir to estimate the potential and cost for hydrogen storage.

Using lakes and reservoirs for hydrogen storage presents substantial policy implications at national, regional, and global levels. At the national level, governments can leverage this innovative storage solution to increase their seasonal energy storage and enhance their energy security and sustainability goals. This would require regulatory frameworks to set standards and ensure the safe and environmentally sound implementation of hydrogen storage in lakes and reservoirs. To accelerate the adoption of the technology, countries can provide subsidies and tax incentives for research, environmental licensing, planning, construction, and maintenance of such technologies. Regionally, hydrogen storage in lakes and reservoirs can facilitate the development of a more resilient and integrated energy grid. Policies encouraging regional cooperation can help create a network of hydrogen storage sites, enhance energy integration between countries, support renewable energy generation, address intermittency issues associated with wind and solar power, and reduce reliance on fossil fuels. It would be wise to consider the storage of hydrogen in lakes and hydropower plants before designing future regional hydrogen pipelines. Globally, the technology aligns with global efforts to transition to clean energy, offering a scalable and cost-effective solution for storing excess renewable energy. International policies, such as those advocated by the United Nations Framework Convention on Climate Change (UNFCCC), can include hydrogen storage as a key strategy for achieving net-zero emissions. Global cooperation is crucial, and policies fostering technology transfer and capacity-building can help developing countries implement hydrogen storage, enhancing their energy security and contributing to global climate goals.

## Methods
The methodology proposed for hydrogen storage in lakes, hydropower, and pumped storage reservoirs is described in Fig. 4a–c. This is possible because hydrogen is insoluble in water and not toxic[25,39]. Hydrogen has even been shown to be beneficial for aquatic environments[40]. In case of leaks or accidents, the hydrogen will rapidly rise in the water columns and dissipate in the atmosphere, not causing harm to the lake, hydropower, or pumped storage reservoir. The hydrogen tank is made of HDPE, which is cheap, easy to handle, and has a low hydrogen permeation[41]. The permeability coefficient of $H_2$ through HPDE (PE100) is $10^{-15}$ mol m m$^{-2}$ s$^{-1}$ Pa$^{-1}$, which is small and can be neglected[41]. Another advantage of this storage system is that the pressure inside the tank should be constant and equal to the hydraulic head of the water column above the tank. This reduces impacts that

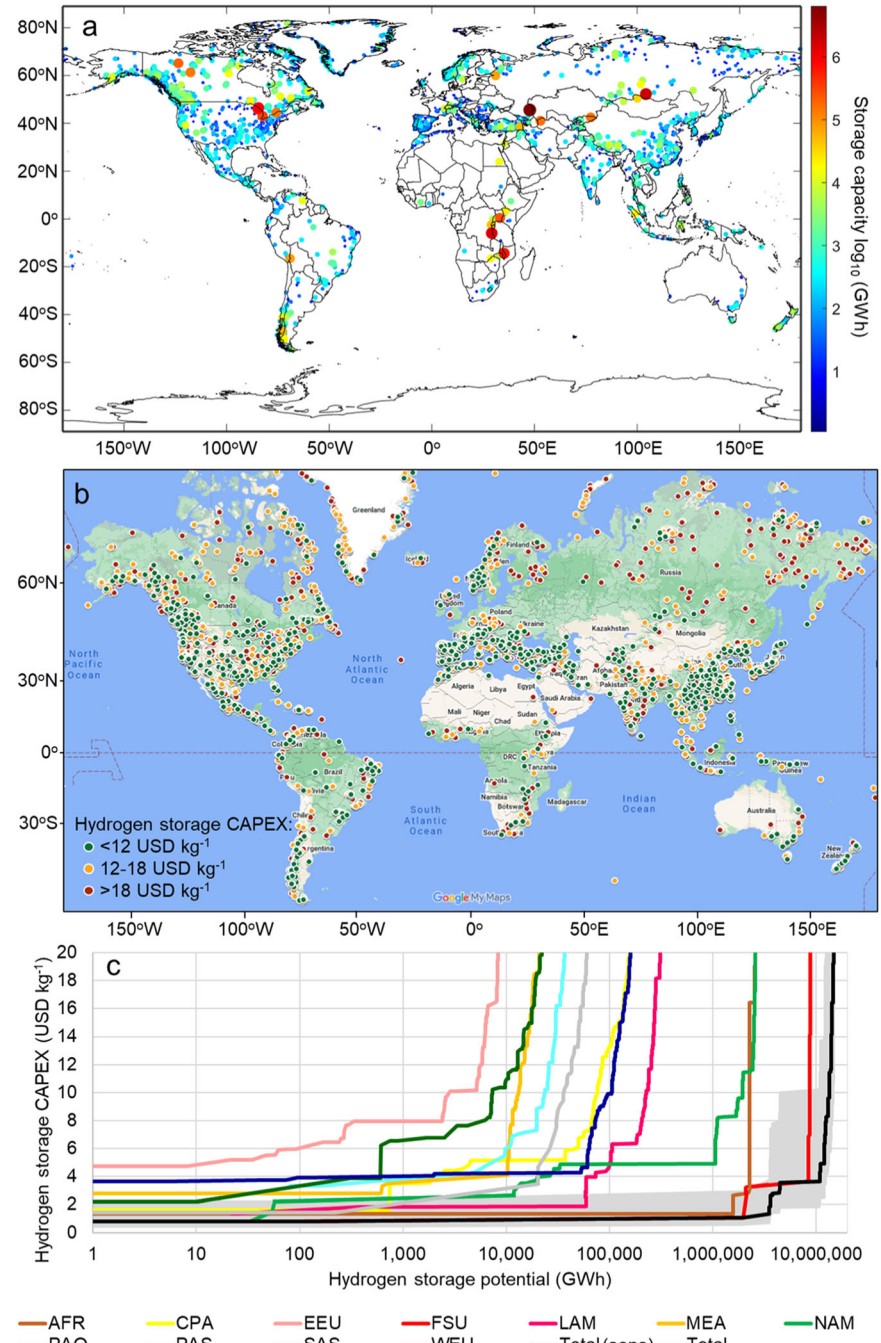

**Fig. 3 | Lakes and reservoirs hydrogen storage global potential. a** storage capacity, (**b**) hydrogen storage CAPEX[32], (**c**) cost-curve per region and global cost-curve, including sensitivity analysis (AFR is Sub-Saharan Africa, WEU is Western Europe, CPA is Centrally Planned Asia and China, EEU is Eastern Europe, FSU is Former Soviet Union, LAM is Latin America and the Caribbean, MEA is Middle East and North Africa, NAM is North America, PAO is Pacific OECD, PAS is Other Pacific Asia, SAS is South Asia).

happen in conventional tanks that suffer large pressure variations, such as fatigue, and increase in permeability and diffusivity. This increases the number of injection/withdrawal cycles, which in turn increases the durability and feasibility of the system. Other aspects involving the use of HDPE concern its life cycle analysis and long-term sustainability. Even though HDPE stores carbon in its composition, which has the potential to reduce atmospheric $CO_2$ concentration (assuming the petrochemical industry uses biomass or direct air capture as a source of carbon), its production and disposal can contribute to ozone depletion, acidification potential, Smog formation, and eutrophication[42].

Gravel is added to the tank to increase its weight and keep the tank at the bottom of the lake. Hydrogen does not react with the gravel in the tanks[25]. A small quantity of the hydrogen stored could react with the dissolved $CO_2$ in the water to produce $CH_4$ through the biological Sabatier reaction[43]. The tank is filled with hydrogen from the top of the reservoir, and water is removed from the bottom, as shown in Fig. 4d. On the other hand, when hydrogen is removed from the tank, water is added to the tank (Fig. 4e). The injection and withdrawal of hydrogen and water from the storage tank must happen simultaneously so that the pressure inside the tank is always the same as the outside pressure. The flow of water in and out is controlled by two

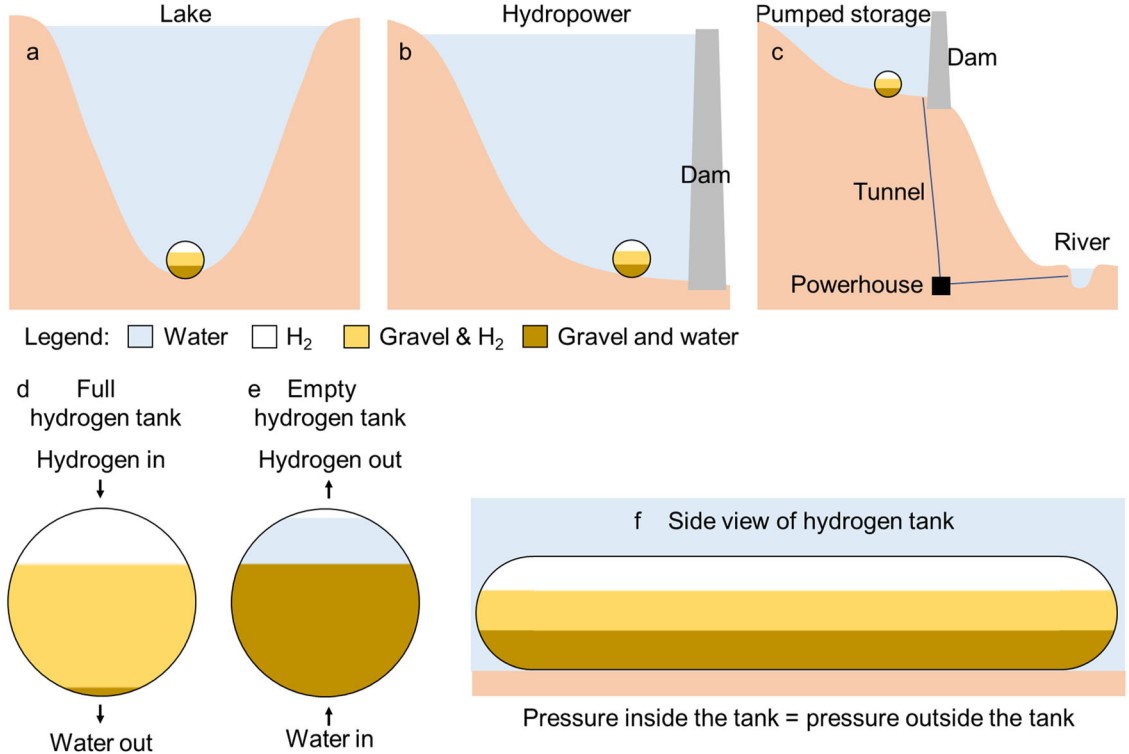

**Fig. 4 | Hydrogen storage. a** lake, (**b**) hydropower and **c** pumped storage reservoirs, (**d**) full hydrogen tank, (**e**) empty hydrogen tank, and (**f**) side view of the hydrogen tank.

analog pressure relief valves one injects water into the tank when the pressure of the tank is lower than its surroundings, and the other withdraws water from the tank when the pressure inside the tank is higher than the surroundings. Mixing water and hydrogen is only an option because the solubility of hydrogen in water is low. Mole fraction solubility varies from 0.00004 to 0.0009 $xH_2(\%)$ at 0 °C at pressures ranging from 1 bar to 50 bar, respectively[25]. Compared with other energy storage solutions, such as batteries and pumped hydro storage, that result in up to 10 and 30% energy losses, respectively. A hydrogen dissolution in water and subsequent loss of 0.0009% per storage cycle can be considered negligible. Figure 4f presents a side view of the hydrogen tank.

The level of hydrogen in the tanks is impacted by three variables: (i) the mass of hydrogen in the tank, (ii) the temperature, and (iii) the pressure at the bottom of the lake or reservoir. The operator can control the mass of hydrogen in the tank by adding or removing hydrogen. The temperature of lakes and hydropower reservoirs below 40 m depth are usually the same throughout the year[33]. The pressure in the tank will vary with the water level of the lake or reservoir. The higher the water level of the reservoir, the higher the pressure in the tanks. The storage capacity of the hydrogen tank fixed to the bottom of the reservoir will vary substantially with the tank's pressure. For example, if the reservoir water level varies from 200 m to 150 m, the pressure in the tank will vary from 20.6 to 15.7 bar. This increases the hydrogen volume in the tank by 24%, assuming that the mass of hydrogen in the tank is kept constant. This is usually not a problem because the hydropower and pumped storage reservoirs, and the hydrogen tanks fill up when electricity is cheap and empty when electricity is expensive. This, however, might not be the case in locations with energy and water conflicts[44].

Equation 1 is used to calculate the pressure in the hydrogen tank. Where $P_T$ is the pressure of the hydrogen tank (in the bar), $P_r$ is the pressure of the atmosphere on the top of the reservoir/lake (assumed to be 1 bar), $D$ is the depth of the storage tank (in m).

The denominator equal to 10.2 is the head of water (m) required to increase the tank's pressure by 1 bar.

$$P_T = P_r + \frac{D}{10.2} \quad (1)$$

To keep the hydrogen tank in the bottom of the lake/reservoir, weight must be added to compensate for the low density of the compressed hydrogen. Gravel was selected the most appropriate material to counteract the buoyant potential of hydrogen. Equation 2 estimates the amount of gravel that must be added to the hydrogen tank to avoid it floating.

$$V \times \rho_W < V_S \times \rho_S + V_W \times \rho_W + V_H \times \rho_H + V_M \times \rho_M \quad (2)$$

Where $V$ is the tank's volume, $\rho_W$ is the water density, $V_S$ is the gravel volume (assuming only the solid part and that the gravel has a 40% porosity), $\rho_S$ is the gravel density, which is assumed to be 2666 kg m$^{-3}$, $V_W$ represents the water volume, $\rho_{SW}$ is the water density, equal to 1000 kg m$^{-3}$, $V_H$ is the maximum hydrogen volume, $\rho_H$ is the hydrogen density, which varies with depth and assuming 15 °C, $V_M$ is the HDPE volume in the pipeline, assuming a 10 cm thickness, $\rho_M$ is the HDPE pipeline density, assumed to be 945 kg m$^{-3}$. As the density of HDPE is similar to the density of water (1000 kg m$^{-3}$), even though the mass of the pipeline of each tank is 300.000 kg, the pipeline has a small contribution to the hydrostatic calculations.

## Data availability

The global potential for $H_2$ storage in lakes and reservoirs open access data generated in this study have been deposited in the figshare database under accession code https://doi.org/10.6084/m9.figshare.26370556.v1. An interactive map with the data is available at the following link: https://www.google.com/maps/d/u/0/edit?mid=1EcGvl_Cr-1l-7WDTw9u4Gvjrx4XnHEo&usp=sharing.

## Code availability

The code is available from the corresponding author upon request.

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

## Acknowledgements
We would like to acknowledge funding for open access from the King Abdullah University of Science and Technology. The contribution of BZ was in part supported by the Austrian Federal Ministry for Climate Action, Environment, Energy, Mobility, Innovation, and Technology (BMK) under the endowed professorship for "Data-Driven Knowledge Generation: Climate Action".

## Author contributions
J.H. conceived the research idea, designed the methodology, and conducted the hydrogen storage cost analysis. A.N. contributed to the assessment of environmental impacts and the sustainability aspects of the hydrogen storage system. O.R. contributed to the global potential estimation and created the interactive map. B.Z. assisted in the hydrogen economy context analysis and provided insights on seasonal storage technologies. J.J., P.D., and T.S. performed the simulations and data analysis for hydrogen storage in lakes and reservoirs. B.Đ. contributed to the economic analysis and feasibility study of the proposed storage system. W.F. conducted the case study on Oroville Lake. M.F. contributed to the hydropower integration aspects and the technical feasibility of the storage system. Y.W. provided critical feedback on the concept, methodology, and data sources. All authors reviewed and approved the final manuscript.

## Competing interests
The authors declare no competing interests in this paper.
