## [Peer Review File · Nature Communications]

REVIEWER COMMENTS

Reviewer #1 (Remarks to the Author):

The manuscript (NCOMMS-24-09690) presents work on “Hydrogen storage in lakes and reservoirs”. The authors discussed varieties of aspects of green hydrogen storage specifically in offshores. The authors nicely explained the whole study and critically examined each scientific corner of the topic, which is interesting and useful in hydrogen and related energy storage industries. The global search for energy storage solutions emphasizes the need for sustainability, cost-effectiveness, and efficiency, with a key focus on reducing carbon footprints compared to current storage technologies. Encouragingly, scientists and researchers are innovating with novel approaches that hold promise in meeting these criteria. Nonetheless, neglecting a comprehensive assessment cradle to the grave may yield illusive outcomes; thus, all the scientific corners must be considered. However, in this study, proper reference works/experiments/studies have been cited to consider a variety of scientific corners related to this approach. The work looks standard but it still requires additional calculations. Investigating the verities of aspects of this approach would be a great addition. In its current format, more attention is given to the implementation. Once a detailed analysis is done on the required scientific corners of the approach/technology (including the operation), it can be considerably suitable for its possible publication in Nature Communications. However, the title of the study seems generalized against the content/work; also, some considerations should be taken by the authors:

Note: The concept/approach presented in this study has already been discussed/published in previous articles (by the same team); for example, <https://doi.org/10.3390/en16073118> and <https://doi.org/10.1016/j.energy.2022.123660>; So this study is a continuation of the previous study with a case study.

General Comments:

1. How the injection and withdrawal (pressurization and depressurization) of hydrogen affect the permeability and diffusivity of HDPE.
2. How long will it be durable (durability: injection-ideal-withdrawal number of cycles)? In the context of the feasibility of the system for a longer period.
3. Sustainability (LCA) of the HDPE (Global warming potential, Ozone depletion potential, Acidification potential, Smog formation potential, Eutrophication potential).
4. The leaked hydrogen will not affect the aquatic life, but the extent of leakage (in a long run) may hamper the efficiency and both the CAPEX and OPEX.

Specific Comments:

1. Why the weight of the HDPE was ignored while balancing the buoyancy force in equation (3)? What should be the preferable thickness of the storage tank and how does it affect the hydrostatic calculations?
2. The density of the HDPE ranges from 930-970 (kg/m³). For a 100m long and 10m diameter pipe or storage tank will require around 150,000 kg (1500000 N; weight) of material (HDPE) just for a thickness of 0.1m and it becomes 1,500,000 kg (15000000 N; weight) if the thickness is 0.5m. I may have a wrong calculation, so please double-check if it is a concern. How do the authors see the

LCA of this huge amount of material specifically the carbon footprint?

3. Initiating the injection of water from the bottom and the withdrawal of hydrogen from the top simultaneously is essential. Due to the substantial difference in the density of these two fluids, even at a depth of 1000m, where the contrast is significant (100 times), ensuring the stability (metacentric) of the tank becomes crucial. How can the control and management of the fluctuating intake and outflow of both fluids be effectively achieved?
4. If filling the tank with sand is required to just increase the weight and keep the tank sink, it would be great to plot the minimum sand requirement (mass or volume) against varying volume and/or dimensions of the tank, Also, a discussion on stored hydrogen volume in the same context.
5. Calculation of stresses and burst pressure against the safety criteria would be interesting.
6. Plotting the rise in the water level for different dimensions of the lake, and discussion on suitability against the size of the lakes, rivers, and other water systems.
7. The cost of storing per unit of hydrogen/energy through this approach is not clear in its current format. Why, around 65% volume was filled with sand if filling the tank with sand is required to just increase the weight and keep the tank sink? If it is loosely filled, how the variation in porosity does affect the process, if any? Capillarity effect during different cycles and its effect on storage volume?

Reviewer #2 (Remarks to the Author):

The authors proposed an alternative for hydrogen storage in lakes and water reservoirs and concluded that it provides cheap and abundant hydrogen storage. The work does not involve any experimentation and is of a theoretical nature only. The article merits publication, but not before addressing the following queries:

-Perhaps my biggest concern is that lakes and reservoirs collect water from springs, etc. and help in avoiding overflowing (flooding) water. Now as the authors suggested to use a 10m diameter pipe in the basin for an average depth of 30m water body, and they suggested to have many of these pipes in the basin. In this way, a large portion of reservoir would be covered by pipes (sand and hydrogen) and the chances of flooding would increase many times. How can the authors address this issue in their proposed scheme?

-The authors mentioned the cost of storing H₂ in HDPE pipe to be 3.76 USD/kg H₂ and argued that this cost is far less than storing H₂ in salt caverns, etc. The authors calculated only the capital investment to make their point and ignored the cost of the installation of this facility. Additionally, why this capital cost is lower than the capital cost involved in storing H₂ in natural (already available) cavities, which may not require any capital investment or only a small one for their formation.

-In the introduction section, the authors talked about the European Union renewable energy scenario, but not that of the USA or the overall world. However, they conducted their case-study in the USA. For conformity, it would be good to talk about the renewable energy scenario and H₂ storage, at least, in the USA.

-Heading "Global Potential" appeared twice.

-Some software mistakes at line 155 and line 220.

Response to reviewers

Hydrogen storage with sand and pipes in lakes and reservoirs

Reviewer #1:

Comment	Response
The manuscript (NCOMMS-24-09690) presents work on “Hydrogen storage in lakes and reservoirs”. The authors discussed varieties of aspects of green hydrogen storage specifically in offshores. The authors nicely explained the whole study and critically examined each scientific corner of the topic, which is interesting and useful in hydrogen and related energy storage industries. The global search for energy storage solutions emphasizes the need for sustainability, cost-effectiveness, and efficiency, with a key focus on reducing carbon footprints compared to current storage technologies. Encouragingly, scientists and researchers are innovating with novel approaches that hold promise in meeting these criteria. Nonetheless, neglecting a comprehensive assessment cradle to the grave may yield illusive outcomes; thus, all the scientific corners must be considered. However, in this study, proper reference works/experiments/studies have been cited to consider a variety of scientific corners related to this approach. The work looks standard but it still requires additional calculations. Investigating the verities of aspects of this approach would be a great addition. In its current format, more attention is given to the implementation. Once a detailed analysis is done on the required scientific corners of the approach/technology (including the operation), it can be considerably suitable for its possible publication in Nature Communications.	Thanks for your positive feedback and valuable contributions to the paper.
However, the title of the study seems generalized against the content/work; also, some considerations should be taken by the authors:	As requested, we changed the title of the paper to “Hydrogen storage with sand and pipes in lakes and reservoirs”.
Note: The concept/approach presented in this study has already been discussed/published in previous articles (by the same team); for example, https://doi.org/10.3390/en16073118 and https://doi.org/10.1016/j.energy.2022.123660; So this study is a continuation of the previous study with a case study.	Thanks for mentioning this. Correct. These papers propose the use sand in pipelines to store gases in the deep ocean. Many papers propose technologies to store compressed air in the deep ocean. However, we could not find a paper that proposed the storage of hydrogen in reservoirs and lakes. We understand this is not a huge contribution, but it is a relevant contribution to the literature.
General Comments: 1. How the injection and withdrawal (pressurization and depressurization) of hydrogen affect the permeability and diffusivity of HDPE.	Good point. We added: “Another advantage of this storage system is that the pressure inside the tank should be constant and equal to the hydraulic head of the water column above the tank. This

	reduces the impacts that happen in conventional tanks that suffer large pressure variations, such as fatigue, and increase in permeability and diffusivity.”
2. How long will it be durable (durability: injection-ideal-withdrawal number of cycles)? In the context of the feasibility of the system for a longer period.	We added: “This increases the number of injection/withdrawal cycles, which in turn increases the durability and feasibility of the system”.
3. Sustainability (LCA) of the HDPE (Global warming potential, Ozone depletion potential, Acidification potential, Smog formation potential, Eutrophication potential).	We added: “Other aspects involving the use of HDPE concern its life cycle analysis and long-term sustainability. Even though HDPE stores carbon in its composition, which has the potential for reducing atmospheric CO ₂ concentration (assuming the petrochemical industry uses biomass or direct air capture as a source of carbon), its production and disposal can contribute to ozone depletion, acidification potential, Smog formation and eutrophication ³⁸ .”
4. The leaked hydrogen will not affect the aquatic life, but the extent of leakage (in a long run) may hamper the efficiency and both the CAPEX and OPEX.	We added: “Compared with other energy storage solutions, such as batteries and pumped hydro storage, that result in up to 10 and 30% energy losses, respectively. A hydrogen dissolution in water and subsequent loss of 0.0009% per storage cycle can be considered negligible.”
Specific Comments: 1. Why the weight of the HDPE was ignored while balancing the buoyancy force in equation (3)? What should be the preferable thickness of the storage tank and how does it affect the hydrostatic calculations? 2. The density of the HDPE ranges from 930-970 (kg/m³). For a 100m long and 10m diameter pipe or storage tank will require around 150,000 kg (1500000 N; weight) of material (HDPE) just for a thickness of 0.1m and it becomes 1,500,000 kg (15000000 N; weight) if the thickness is 0.5m. I may have a wrong calculation, so please double-check if it is a concern.	We changed Equation 3 to better represent the pipeline contribution to the hydrostatic calculations. Thanks for sharing these calculations. 10 cm thick is a good estimate for a robust pipeline that can sustain the pressure caused by the sand inside. We have double-checked the calculation and got a 300,000 kg estimate for a 10 cm thickness. You may have added the thickness once, but it should be counted in the two extremities of the pipe. We added: “V_M is the HDPE volume in the pipeline, assuming a 10 cm thickness, and ρ_M is the HDPE pipeline density, assumed to be 945 kg/m³. As the density of HDPE is similar to the density of water (1000 kg/m³), even though the mass of the pipeline of each tank is 300.000 kg, the pipeline has a small contribution to the hydrostatic calculations”.
How do the authors see the LCA of this huge amount of material specifically the carbon footprint?	In the future, carbon-based materials will provide an opportunity to store carbon extracted from the atmosphere. However, this would require the petrochemical industry to use biomass or direct air capture as a carbon source. We added: “Even though HDPE stores carbon on its composition, which has the potential to reduce atmospheric CO₂ concentration (assuming the petrochemical industry uses biomass or direct air capture as a source of carbon),”
3. Initiating the injection of water from the bottom and the withdrawal of hydrogen from the top simultaneously is	We added: “The injection and withdrawal of hydrogen and water from the storage tank must

essential. Due to the substantial difference in the density of these two fluids, even at a depth of 1000m, where the contrast is significant (100 times), ensuring the stability (metacentric) of the tank becomes crucial. How can the control and management of the fluctuating intake and outflow of both fluids be effectively achieved?	happen simultaneously so that the pressure inside the tank is always the same as the outside pressure. The flow of water in and out is controlled by two analog pressure relief valves, one injects water into the tank when the pressure of the tank is lower than the surrounding, and the other withdraws water from the tank when the pressure inside the tank is higher than the surroundings. Note that there must be a slightly higher hydrogen pressure in the tank (around 0.25 bar⁴⁰) for the withdrawal of water from the tank to overcome the capillary action of water in sand.” Metacentric stability does not apply in this case because the pipeline is designed not to float in any circumstance.
4. If filling the tank with sand is required to just increase the weight and keep the tank sink, it would be great to plot the minimum sand requirement (mass or volume) against varying volume and/or dimensions of the tank, Also, a discussion on stored hydrogen volume in the same context.	Thanks for this contribution. We added Fig. 1d “d H₂ and sand volume required to avoid the tank from submerging”. We also added: “Fig. 1d shows that the maximum hydrogen volume that can be stored in the tank without submerging is 62.53% of the total volume at a 200 meters depth, including the void between the sand particles”.
5. Calculation of stresses and burst pressure against the safety criteria would be interesting.	We cannot estimate the burst pressure and stresses for the proposed tanks. This is because the tanks will never burst, given that the pressure inside the tank is always the same as the pressure outside the tank.
6. Plotting the rise in the water level for different dimensions of the lake, and discussion on suitability against the size of the lakes, rivers, and other water systems.	We added: “It should be noted that the hydrogen storage tanks should only be installed below the reservoir's dead storage capacity or the lake's minimum historical level. If the tank ends up above the water level, it cannot be used for hydrogen storage, and the tanks will reduce the water storage capacity of the reservoir or lake and their capacity to control drought and floods.” We added: “Note that 30 meters depth is too shallow for building hydrogen storage tanks, as the pressure will be only approximately 4 bar and the cost for storing hydrogen will be high as shown in Fig. 1d. We added reservoirs with 30 meters average depth to the global potential methodology because 10% of the total reservoir area of the reservoir might achieve 100 meters depth or more, which is required for underwater hydrogen store. However, this might not be the case for all reservoirs. This is a limitation of the methodology and data available. For a precise estimate of the hydrogen storage cost and potential of the reservoir, the reader needs to find bathymetric data of the reservoir and use the equations in the paper to estimate the potential and cost for hydrogen storage.”

7. The cost of storing per unit of hydrogen/energy through this approach is not clear in its current format. Why, around 65% volume was filled with sand if filling the tank with sand is required to just increase the weight and keep the tank sink? If it is loosely filled, how the variation in porosity does affect the process, if any?	We added: “Sand was selected as ballast to increase the hydrogen tank's weight and keep it in the bottom of the reservoir. This is because sand is the cheapest material available for this service, costing as low as 1 USD/ton. The pipeline could be filled with concrete, but concrete degrades with time (sand does not degrade with time) and the cost of concrete is around 30 USD/ton, significantly increasing the cost of hydrogen storage. The main characteristic of sand that impacts hydrogen storage costs is its density. The larger the density, the less sand mass is required, and thus, more hydrogen can be stored in the tank. The sand porosity does not impact the total hydrogen stored in the tank. It impacts the capillarity of water, which is resolved with a slight increase in hydrogen pressure in the tank (around 0.25 bar⁴⁰).”
Capillarity effect during different cycles and its effect on storage volume?	We added: “Note that there must be a slightly higher hydrogen pressure in the tank (around 0.25 bar⁴⁰) for the withdrawal of water from the tank to overcome the capillary action of water in sand.”

Reviewer #2:

Comment	Response
The authors proposed an alternative for hydrogen storage in lakes and water reservoirs and concluded that it provides cheap and abundant hydrogen storage. The work does not involve any experimentation and is of a theoretical nature only. The article merits publication, but not before addressing the following queries:	Thanks for your positive feedback and valuable contributions to the paper.
-Perhaps my biggest concern is that lakes and reservoirs collect water from springs, etc. and help in avoiding overflowing (flooding) water. Now as the authors suggested to use a 10m diameter pipe in the basin for an average depth of 30m water body, and they suggested to have many of these pipes in the basin. In this way, a large portion of reservoir would be covered by pipes (sand and hydrogen) and the chances of flooding would increase many times. How can the authors address this issue in their proposed scheme?	Thanks for sharing the reader's perspective. We removed “average depth threshold of 30 meters” and added “specifications described in the methods section”. We did this because we did not want to confuse the reader that H₂ should be stored in 30 meters depth tanks. This is too shallow for storing H₂ for the reasons you mentioned and because of the high costs involved. We added: “Note that 30 meters depth is too shallow for building hydrogen storage tanks, as the pressure will be only approximately 4 bar and the cost for storing hydrogen will be high as shown in Fig. 1d. We added reservoirs with 30 meters average depth to the global potential methodology because 10% of the total reservoir area of the reservoir might achieve 100 meters depth or more, which is required for underwater hydrogen store. However, this might not be the case for all reservoirs. This is a limitation of the methodology

	and data available. For a precise estimate of the hydrogen storage cost and potential of the reservoir, the reader needs to find bathymetric data of the reservoir and use the equations in the paper to estimate the potential and cost for hydrogen storage.” We also added: “It should be noted that the hydrogen storage tanks should only be installed below the reservoir's dead storage capacity or the lake's minimum historical level. If the tank ends up above the water level, it cannot be used for hydrogen storage, and the tanks will reduce the water storage capacity of the reservoir or lake and their capacity to control drought and floods”.
-The authors mentioned the cost of storing H2 in HDPE pipe to be 3.76 USD/kg H2 and argued that this cost is far less than storing H2 in salt caverns, etc. The authors calculated only the capital investment to make their point and ignored the cost of the installation of this facility. Additionally, why this capital cost is lower than the capital cost involved in storing H2 in natural (already available) cavities, which may not require any capital investment or only a small one for their formation.	Thanks for pointing this out. We added: “5 to 25 times cheaper than storing hydrogen in newly built salt caverns³⁹”. We also added: “However, if already available, preference should be given to storing hydrogen in natural cavities, salt caverns and depleted natural gas reservoirs, as they require lower capital investment and might have higher storage capacities”. Regarding the installation costs, please see Supplementary Table 2. We assume a 50% construction/installation cost. This high because it involves underwater construction.
-In the introduction section, the authors talked about the European Union renewable energy scenario, but not that of the USA or the overall world. However, they conducted their case-study in the USA. For conformity, it would be good to talk about the renewable energy scenario and H2 storage, at least, in the USA.	We added: “In the USA, renewable generation in 2023 was 21%, with solar power expected to increase by 30% between 2023 and 2024⁴”. We added: “According to McKinsey & Company¹⁴, the demand for grey hydrogen was 100 million tons in 2023 and estimates that 600 million tons of green hydrogen will be required by 2050 to achieve net zero emissions.”
-Heading "Global Potential" appeared twice.	Thanks for spotting this. We removed the heading “Global potential” from the Methods sections.
-Some software mistakes at line 155 and line 220.	Thanks for sharing this. We fixed the problem.

REVIEWER COMMENTS

Reviewer #1 (Remarks to the Author):

The suggestions raised in the last review stage have now been considered by the authors and updated in the revised draft. However, some points are still not clearly defined/discussed (raised in the previous version of this draft).

R1- How do the authors see the LCA of this huge amount of material specifically the carbon footprint?

Authors: "Even though HDPE stores carbon in its composition, which has the potential to reduce atmospheric CO₂ concentration."

R1- If the exposed surface of the material is submerged in water, how will it reduce atmospheric CO₂ concentration? It might reduce a portion of the dissolved CO₂, I guess. Even if it does reduce atmospheric CO₂, how long would it take to offset the carbon emitted during the manufacturing of this particular amount of HDPE? Highlighting the carbon emissions range (or LCA) involved in manufacturing the required HDPE material would be greatly beneficial.

Regarding General Comment #2: Initially, before injecting hydrogen, the sand system is water-wet. During the first cycle of hydrogen injection, capillarity will play a significant role. In subsequent cycles, permeability hysteresis might be observed. It would be beneficial to include a discussion of these laboratory observations in the draft, particularly if such behavior was noticed during the experiment. Additionally, if permeability hysteresis is present, could it pose difficulties in regulating the pressure and flow (in and out, of the system) in each cycle? This aspect should be explored to understand its impact on the experimental outcomes.

I find the comparison of sand with concrete in the cost analysis to be unnecessary. Instead, the overall cost, particularly the cost per unit of hydrogen through this storage process, should be compared with other common and available hydrogen storage methods. A comparative table would be beneficial, including short-term storage systems like hydrogen tanks, material-based storage, and salt caverns; as well as, bulk storage systems such as hydrogen storage in aquifers or reservoirs. This approach will provide a clearer and more relevant cost comparison across different storage methods.

Reviewer #2 (Remarks to the Author):

The authors have answered all my queries. I am satisfied, and I recommend the publication of the manuscript.

Response to referees

Hydrogen storage with gravel and pipes in lakes and reservoirs

Reviewer #1:

Comment	Response
The suggestions raised in the last review stage have now been considered by the authors and updated in the revised draft. However, some points are still not clearly defined/discussed (raised in the previous version of this draft). R1- How do the authors see the LCA of this huge amount of material specifically the carbon footprint? Authors: "Even though HDPE stores carbon in its composition, which has the potential to reduce atmospheric CO₂ concentration." R1- If the exposed surface of the material is submerged in water, how will it reduce atmospheric CO₂ concentration? It might reduce a portion of the dissolved CO₂, I guess. Even if it does reduce atmospheric CO₂, how long would it take to offset the carbon emitted during the manufacturing of this particular amount of HDPE? Highlighting the carbon emissions range (or LCA) involved in manufacturing the required HDPE material would be greatly beneficial.	We added: The cradle-to-gate life cycle analysis for producing HDPE using fossil fuels as materials and energy sources result in an overall CO₂ emission of 1.6 kg per kg of HDPE [2]. Thus, a 300 tons tank would emit 480 tons of CO₂. A gas power plant to generate 186 MWh (same energy stored as H₂ in the tank) would be 77 tons, assuming an emission of 413 kg of CO₂/MWh [3]. If the tank operates seasonally, it would take 6.2 Years for the tank to store the same amount of energy per CO₂ emission as a gas power plant. In the future, with the use of renewable electricity and CO₂ captured from the air, the production of the tank with the methanol-to-olefins process would result in negative emissions [1]. For instance, 85.7% of the tank's mass is carbon, which is equivalent to 257 tons of carbon or 942 tons of CO₂ emissions captured and stored within the HDPE tank.
Regarding General Comment #2: Initially, before injecting hydrogen, the sand system is water-wet. During the first cycle of hydrogen injection, capillarity will play a significant role. In subsequent cycles, permeability hysteresis might be observed. It would be beneficial to include a discussion of these laboratory observations in the draft, particularly if such behavior was noticed during the experiment. Additionally, if permeability hysteresis is present, could it pose difficulties in regulating the pressure and flow (in and out, of the system) in each cycle? This aspect should be explored to understand its impact on the experimental outcomes.	Thanks for pointing out these challenges for injecting and extracting hydrogen from the tank. Unfortunately, our research group lacks the funding and equipment to perform the proposed lab experiments. To avoid capillarity and permeability hysteresis we decided to change sand to gravel from mine waste with a granularity above 5cm. We changed the manuscript and the supplementary information accordingly. We added: "Gravel from mine waste with granularity above 5 cm was selected as ballast to increase the hydrogen tank's weight and keep it in the bottom of the reservoir. This is because using material with lower granularity, such as sand, would result in challenges to insert the H₂ in the tank due to the capillary effect of the water and permeability hysteresis when removing the H₂ from the tank. The cost of the gravel from mine waste varies depending on the availability of mining activity near the hydrogen storage location and the transport distance from the source. Finding other uses for mine waste is convenient as it reduces the costs for discarding it."
I find the comparison of sand with concrete in the cost analysis to be unnecessary. Instead, the overall cost, particularly the cost per unit of hydrogen through this storage process, should be compared with other common and available hydrogen storage methods. A comparative table would be beneficial, including short-term storage systems like hydrogen tanks, material-based storage, and salt caverns; as well as, bulk	We added Table 2 to the manuscript. We also added the levelized cost of hydrogen storage to the paper. We could not find cost estimates for material-based hydrogen storage.

storage systems such as hydrogen storage in aquifers or reservoirs. This approach will provide a clearer and more relevant cost comparison across different storage methods.	
--	--

Reviewer #2:

Comment	Response
The authors have answered all my queries. I am satisfied, and I recommend the publication of the manuscript.	Thanks again for your valuable contributions to the paper.

REVIEWERS' COMMENTS

Reviewer #1 (Remarks to the Author):

I am satisfied with the revised draft and appreciate the author's careful consideration of most of the suggestions I raised. The revised draft now appears scientifically robust and can be considered for publication. However, I recommend double-checking the draft, as 'sand' has been replaced with 'gravel' throughout the text. This change requires careful modification in various sections, including the text, figures, and tables. Additionally, if applicable, please specify 'cleaned gravel' instead of 'gravel' once at the beginning.

Response to referees

Hydrogen storage with gravel and pipes in lakes and reservoirs

Reviewer #1:

Comment	Response
I am satisfied with the revised draft and appreciate the author's careful consideration of most of the suggestions I raised. The revised draft now appears scientifically robust and can be considered for publication.	Thanks for your positive feedback and valuable contributions to the paper.
However, I recommend double-checking the draft, as 'sand' has been replaced with 'gravel' throughout the text. This change requires careful modification in various sections, including the text, figures, and tables.	Thanks for pointing this out. We have double-checked the latest draft and replaced the following: 1) Fig 1f: changed 'H ₂ (sand area)' to 'H ₂ (gravel volume)'. 2) 'cheaper than storing hydrogen in salt caverns' with 'competitive with "other large scale hydrogen storage options'. 3) We removed the text below because the capillary action of water in gravel can be neglected. 'Note that there must be a slightly higher hydrogen pressure in the tank (around 0.25 bar ⁴⁷) for the withdrawal of water from the tank to overcome the capillary action of water in gravel.'
Additionally, if applicable, please specify 'cleaned gravel' instead of 'gravel' once at the beginning.	We have specified 'clean gravel' instead of 'gravel' in the phase below: 'Clean gravel from mine waste with granularity above 5 cm was selected as ballast to increase the hydrogen tank's weight and keep it in the bottom of the reservoir.'
	The "Figures.pptx" file is not showing up properly because it is a PowerPoint file and the images are larger than the slide frame.